# Masked Trajectory Models for Prediction, Representation, and Control

**Philipp Wu** [1 2]  **Arjun Majumdar** [† 3]  **Kevin Stone** [† 1]  **Yixin Lin** [† 1]  **Igor Mordatch** [4]

**Pieter Abbeel** [2]  **Aravind Rajeswaran** [1]

## Abstract

We introduce Masked Trajectory Models (MTM) as a generic abstraction for sequential decision making. MTM takes a trajectory, such as a state-action sequence, and aims to reconstruct the trajectory conditioned on random subsets of the same trajectory. By training with a highly randomized masking pattern, MTM learns versatile networks that can take on different roles or capabilities, by simply choosing appropriate masks at inference time. For example, the same MTM network can be used as a forward dynamics model, inverse dynamics model, or even an offline RL agent. Through extensive experiments in several continuous control tasks, we show that the same MTM network – i.e. same weights – can match or outperform specialized networks trained for the aforementioned capabilities. Additionally, we find that state representations learned by MTM can significantly accelerate the learning speed of traditional RL algorithms. Finally, in offline RL benchmarks, we find that MTM is competi-

tive with specialized offline RL algorithms, despite MTM being a generic self-supervised learning method without any explicit RL components. Code is available at https://github.com/facebookresearch/mtm.

## 1. Introduction

Sequential decision making is a field with a long and illustrious history, spanning various disciplines such as reinforcement learning (Sutton & Barto, 1998), control theory (Bertsekas, 1995; Åström & Murray, 2008), and operations research (Powell, 2007). Throughout this history, several paradigms have emerged for training agents that can achieve long-term success in unknown environments. However, many of these paradigms necessitate the learning and integration of multiple component pieces to obtain decision-making policies. For example, model-based RL methods require the learning of world models and actor-critic methods require the learning of critics. This leads to complex and unstable multi-loop training procedures and often requires various ad-hoc stabilization techniques. In parallel, the emergence of self-supervised learning (Devlin et al., 2018; Jing & Tian, 2019) has led to the development of simple training objectives such as masked prediction and contrastive prediction, which can train generic backbone models for various tasks in computer vision and natural language processing (NLP). Motivated by this advancement,

[†]Equal second author contribution, listed alphabetically. [1]Meta AI [2]UC Berkeley [3]Georgia Tech [4]Google Research. Correspondence to: Philipp <philippwu@berkeley.edu>.

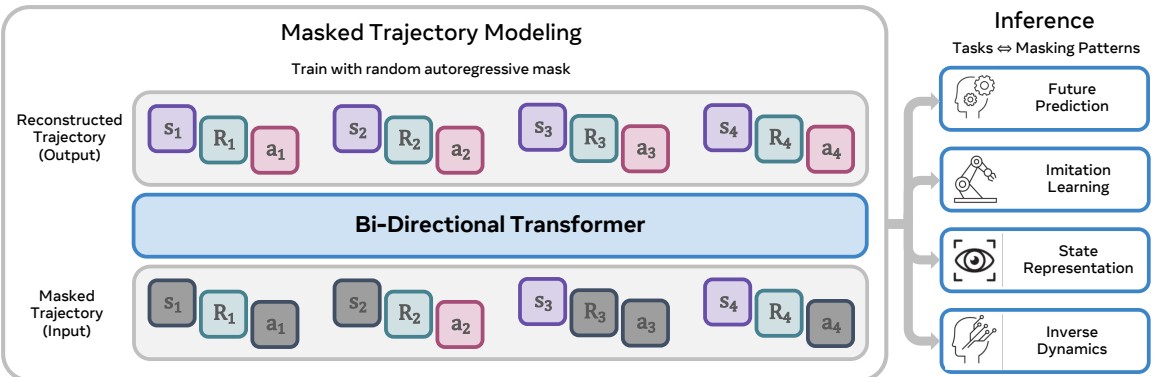

*Figure 1.* **Masked Trajectory Modeling (MTM) Framework.** (Left) The training process involves reconstructing trajectory segments from a randomly masked view of the same. (Right) After training, MTM can enable several downstream use-cases by simply changing the masking pattern at inference time. See Section 3 for discussion on training and inference masking patterns.

we explore if self-supervised learning can lead to the creation of generic and versatile models for sequential decision making with capabilities including future prediction, imitation learning, and representation learning.

Towards this end, we propose the use of Masked Trajectory Models (MTM) as a generic abstraction and framework for prediction, representation, and control. Our approach draws inspiration from two recent trends in Artificial Intelligence. The first is the success of masked prediction, also known as masked autoencoding, as a simple yet effective self-supervised learning objective in NLP (Devlin et al., 2018; Liu et al., 2019; Brown et al., 2020) and computer vision (Bao et al., 2021; He et al., 2021). This task of masked prediction not only forces the model to learn good representations but also develops its conditional generative modeling capabilities. The second trend that inspires our work is the recent success of transformer sequence models, such as decision transformers, for reinforcement (Chen et al., 2021; Janner et al., 2021) and imitation learning (Reed et al., 2022; Shafiullah et al., 2022). Motivated by these breakthroughs, we investigate if the combination of masked prediction and transformer sequence models can serve as a generic self-supervised learning paradigm for decision-making.

Conceptually, MTM is trained to take a trajectory sequence of the form: $\boldsymbol{\tau} := (\mathbf{s}_k, \mathbf{a}_k, \mathbf{s}_{k+1}, \mathbf{a}_{k+1}, \ldots \mathbf{s}_t, \mathbf{a}_t)$ and reconstruct it given a masked view of the same, i.e.

$$\hat{\boldsymbol{\tau}} = \boldsymbol{h}_\theta \left( \texttt{Masked}(\boldsymbol{\tau}) \right) \qquad \text{(MTM)}$$

where $\boldsymbol{h}_\theta(\cdot)$ is a bi-directional transformer and $\texttt{Masked}(\boldsymbol{\tau})$ is a masked view of $\boldsymbol{\tau}$ generated by masking or dropping some elements in the sequence. For example, one masked view of the above sequence could be: $(\mathbf{s}_k, \_\_, \_\_, \mathbf{a}_{k+1}, \_\_, \ldots, \mathbf{s}_t, \_\_)$ where $\_\_$ denotes a masked element. In this case, MTM must infill intermediate states and actions in the trajectory as well as predict the next action in the sequence. A visual illustration of our paradigm is shown in Figure 1. Once trained, MTM can take on multiple roles or capabilities at inference time by appropriate choice of masking patterns. For instance, by unmasking actions and masking states in the sequence, MTM can function as a forward dynamics model.

**Our Contributions** Our main contribution is the proposal of MTM as a versatile modeling paradigm and pre-training method. We empirically investigate the capabilities of MTM on several continuous control tasks including planar locomotion (Fu et al., 2020) and dexterous hand manipulation (Rajeswaran et al., 2018). We highlight key findings and unique capabilities of MTM below.

1. **One Model, Many Capabilities:** The same model trained with MTM (i.e. the same set of weights) can be used zero-shot for multiple purposes including inverse dynamics, forward dynamics, imitation learning, offline RL, and representation learning.

2. **Heteromodality:** MTM is uniquely capable of consuming heteromodal data and performing missing data imputation, since it was trained to reconstruct full trajectories conditioned on randomly masked views. This capability is particularly useful when different trajectories in the dataset contain different modalities, such as a dataset containing both state-only trajectories as well as state-action trajectories (Baker et al., 2022). Following the human heteromodal cortex (Donnelly, 2011), we refer to this capability as heteromodality.

3. **Data Efficiency:** Training with random masks enables different training objectives or combinations, thus allowing more learning signal to be extracted from any given trajectory. As a result, we find MTM to be more data efficient compared to other methods.

4. **Representation Learning:** We find that state representations learned by MTM transfer remarkably well to traditional RL algorithms like TD3 (Fujimoto et al., 2018a), allowing them to quickly reach optimal performance. This suggests that MTM can serve as a powerful self-supervised pre-training paradigm, even for practitioners who prefer to use conventional RL algorithms.

Overall, these results highlight the potential for MTM as a versatile paradigm for RL, and its ability to be used as a tool for improving the performance of traditional RL methods.

## 2. Related Work

**Autoencoders and Masked Prediction.** Autoencoders have found several applications in machine learning. The classical PCA (Jolliffe & Cadima, 2016) can be viewed as a linear autoencoder. Denoising autoencoders (Vincent et al., 2008) learn to reconstruct inputs from noise corrupted versions of the same. Masked autoencoding has found recent success in domains like NLP (Devlin et al., 2018; Brown et al., 2020) and computer vision (He et al., 2021; Bao et al., 2021). Our work explores the use of masked prediction as a self-supervised learning paradigm for RL.

**Offline Learning for Control** Our work primarily studies the offline setting for decision making, where policies are learned from static datasets. This broadly falls under the paradigm of offline RL (Lange et al., 2012). A large class of offline RL algorithms modify their online counterparts by incorporating regularization to guard against distribution shift that stems from the mismatch between offline training and online evaluation (Kumar et al., 2020; Kidambi et al., 2020; Fujimoto et al., 2018b; Yu et al., 2021; Liu et al., 2020). In contrast, our work proposes a generic self-supervised pre-

training paradigm for decision making, where the resulting model can be directly repurposed for offline RL.

**Self-Supervised Learning for Control** The broad idea of self-supervision has been incorporated into RL in two ways. The first is self-supervised **data collection**, such as task-agnostic and reward-free exploration (Pathak et al., 2017; Laskin et al., 2021; Burda et al., 2018). The second is concerned with self-supervised **learning** for control, which is closer to our work. Prior works typically employ self-supervised learning to obtain state representations (Yang & Nachum, 2021; Parisi et al., 2022; Nair et al., 2022; Xiao et al., 2022) or world models (Hafner et al., 2020; Hansen et al., 2022a;b; Seo et al., 2022), for subsequent use in standard RL pipelines. In contrast, MTM uses self-supervised learning to train a single versatile model that can exhibit multiple capabilities.

**Transformers and Attention in RL** Our work is inspired by the recent advances in AI enabled by transformers (Vaswani et al., 2017), especially in offline RL (Chen et al., 2021; Janner et al., 2021; Jiang et al., 2022b) and imitation learning (Reed et al., 2022; Shafiullah et al., 2022; Brohan et al., 2022; Jiang et al., 2022a; Zhou et al., 2022). Of particular relevance are works that utilize transformers in innovative ways beyond the standard RL paradigm. Decision Transformers and related methods (Schmidhuber, 2019; Srivastava et al., 2019; Chen et al., 2021) use return-conditioned imitation learning, which we also adopt in this work. However, in contrast to Chen et al. (2021) and Janner et al. (2021) who use next token prediction as the self-supervised task, we use a bi-directional masked prediction objective. This masking pattern enables the learning of versatile models that can take on different roles based on inference-time masking pattern.

Recently, Liu et al. (2022) and Carroll et al. (2022) explore the use of bi-directional transformers for RL and we build off their work. In contrast to Liu et al. (2022) which studies downstream tasks like goal reaching and skill prompting, we study a different subset of tasks such as forward and inverse dynamics. Liu et al. (2022) also studies offline RL by applying TD3 and modifying the transformer attention mask to be causal, while we study the return conditioned behavior cloning setting. In contrast to Carroll et al. (2022), we study the broader capabilities of our model on several high-dimensional control tasks. VPT (Baker et al., 2022) also tackles sequential decision making using transformers, focusing primarily on extracting action labels with a separate inverse dynamics model. Furthermore, unlike prior work, we also demonstrate that our model has unique and favorable properties like data efficiency, heteromodality, and the capability to learn good state representations.

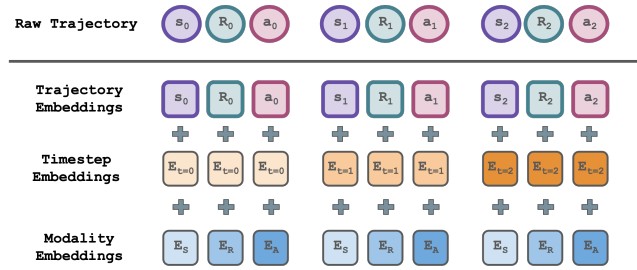

*Figure 2.* **Tokenization of the trajectory sequence** comprises three components. A modality specific encoder lifts from the raw modality space to a common representation space, where we additionally add timestep embeddings and modality type embeddings. Collectively, these allow the transformer to distinguish between different elements in the sequence.

## 3. Masked Trajectory Modeling

We now describe the details of our masked trajectory modeling paradigm, such as the problem formulation, training objective, masking patterns, and overall architecture used.

### 3.1. Trajectory Datasets

MTM is designed to operate on trajectory datasets that we encounter in decision making domains. Taking the example of robotics, a trajectory comprises of proprioceptive states, camera observations, control actions, task/goal commands, and so on. We can denote such a trajectory comprising of $M$ different modalities as

$$\boldsymbol{\tau} = \left\{ \left( \mathbf{x}_1^1, \mathbf{x}_1^2, \ldots \mathbf{x}_1^M \right), \ldots \left( \mathbf{x}_T^1, \mathbf{x}_T^2, \ldots \mathbf{x}_T^M \right) \right\}, \quad (1)$$

where $\mathbf{x}_t^m$ refers to the $m^{\text{th}}$ modality in the $t^{\text{th}}$ timestep. In our empirical investigations, following prior work (Chen et al., 2021; Janner et al., 2021), we use state, action, and return-to-go (RTG) sequences as the different data modalities. Note that in-principle, our mathematical formulation is generic and can handle any modality.

### 3.2. Architecture and Masked Modeling

To perform masked trajectory modeling, we first "tokenize" the different elements in the raw trajectory sequence, by lifting them to a common representation space using modality-specific encoders. Formally, we compute

$$\mathbf{z}_t^m = E_\theta^m(\mathbf{x}_t^m) \quad \forall t \in [1, T], \ m \in [1, M],$$

where $E_\theta^m$ is the encoder corresponding to modality $m$. We subsequently arrange the embeddings in a 1-D sequence of length $N = M \times T$ as:

$$\boldsymbol{\tau} = \left( \mathbf{z}_1^1, \mathbf{z}_1^2, \ldots \mathbf{z}_1^M, \ldots \mathbf{z}_t^m, \ldots \mathbf{z}_T^M \right).$$

The self-supervised learning task in MTM is to reconstruct the above sequence conditioned on a masked view of the same.

We denote the latter with Masked($\boldsymbol{\tau}$), where we randomly drop or "mask" a subset of elements in the sequence. The final self-supervised objective is given by:

$$\max_{\theta} \ \mathbb{E}_{\boldsymbol{\tau}} \sum_{t=1}^{T} \sum_{m=1}^{M} \log P_{\theta}\left(\mathbf{z}_t^m \mid \text{Masked}(\boldsymbol{\tau})\right), \quad (2)$$

where $P_{\theta}$ is the prediction of the model. This encourages the learning of a model that can reconstruct trajectories from parts of it, forcing it to learn about the environment as well as the data generating policy, in addition to good representations of the various modalities present in the trajectory.

**Architecture and Embeddings**   We adopt an encoder-decoder architecture similar to He et al. (2021) and Liu et al. (2022), where both the encoder and decoder are bi-directional transformers. We use a modality-specific encoder to lift the raw trajectory inputs to a common representation space for tokens. Further, to allow the transformer to disambiguate between different elements in the sequence, a fixed sinusoidal timestep encoding and a learnable mode-specific encoding are added, as illustrated in Figure 2. The resulting sequence is then flattened and fed into the transformer encoder where only unmasked tokens are processed. The decoder processes the full trajectory sequence, and uses values from the encoder when available, or a mode-specific mask token when not. The decoder is trained to predict the original sequence, including the unmasked tokens, using an MSE loss (He et al., 2021), which corresponds to a Gaussian probabilistic model. We also note that the length of episodes/trajectories in RL can be arbitrarily long. In our practical implementation, we model shorter "trajectory segments" that are randomly sub-selected contiguous segments of fixed length from the full trajectory.

**Masking Pattern**   Intuitively, we can mask elements in the sequence randomly with a high enough mask ratio to make the self-supervised task difficult. This has found success in computer vision (He et al., 2021). We propose to use a variation of this – a random autoregressive masking pattern. This pattern requires at least one token in the input sequence to be autoregressive, meaning it must be predicted based only on previous tokens, and all future tokens are masked. This means the last element in each sampled trajectory segment is necessarily masked. See Figure 3 for an illustration. We note that the autoregressive mask in our context is **not** using a causal mask in attention weights, but instead corresponds to masking at the input and output token level, similar to MAE.

In the case of computer vision and NLP, the entire image or sentence is often available at inference time. However, in the case of RL, the sequence data is generated as the agent interacts with the environment. As a result, at inference time, the model is forced to be causal (i.e. use only the past tokens).

By using our random autoregressive masking pattern, the model both learns the underlying temporal dependencies in the data, as well as the ability to perform inference on past events. We find that this simple modification is helpful in most tasks we study.

### 3.3. MTM as a generic abstraction for RL

The primary benefit of MTM is its versatility. Once trained, the MTM network can take on different roles, by simply using different masking patterns at inference time. We outline a few examples below. See Figure 3 for a visual illustration.

1. Firstly, MTM can be used as a stand-alone algorithm for offline RL, by utilizing a return-conditioned behavior cloning (RCBC) mask at inference time, analogous to DT (Chen et al., 2021) and RvS (Emmons et al., 2021). However, in contrast to DT and RvS, we use a different self-supervised pre-training task and model architecture. We find in Section 4.3 that using MTM in "RCBC-mode" outperforms DT and RvS.

2. Alternatively, MTM can be used to recover various components that routinely feature in traditional RL pipelines, as illustrated in Figure 3. Conceptually, by appopriate choice of masking patterns, MTM can: (a) provide state representation that accelerates the learning of traditional RL algorithms; (b) perform policy initialization through behavior cloning; (c) act as a world model for model-based RL algorithms; (d) act as an inverse dynamics model to recover action sequences that track desired reference state trajectories.

## 4. Experiments

Through detailed empirical evaluations, we aim to study the following questions.

1. Is MTM an effective algorithm for offline RL?

2. Is MTM a versatile learner? Can the same network trained with MTM be used for different capabilities without additional training?

3. Is MTM an effective heteromodal learner? Can it consume heteromodal datasets, like state-only and state-action trajectories, and effectively use such a dataset to improve performance?

4. Can MTM learn good representations that accelerate downstream learning with standard RL algorithms?

See Appendix for additional details about model architecture and hyperparameters.

### 4.1. Benchmark Datasets

To help answer the aforementioned questions, we draw upon a variety of continuous control tasks and datasets that leverage the MuJoCo simulator (Todorov et al., 2012). Additional environment details can be found in Appendix B.

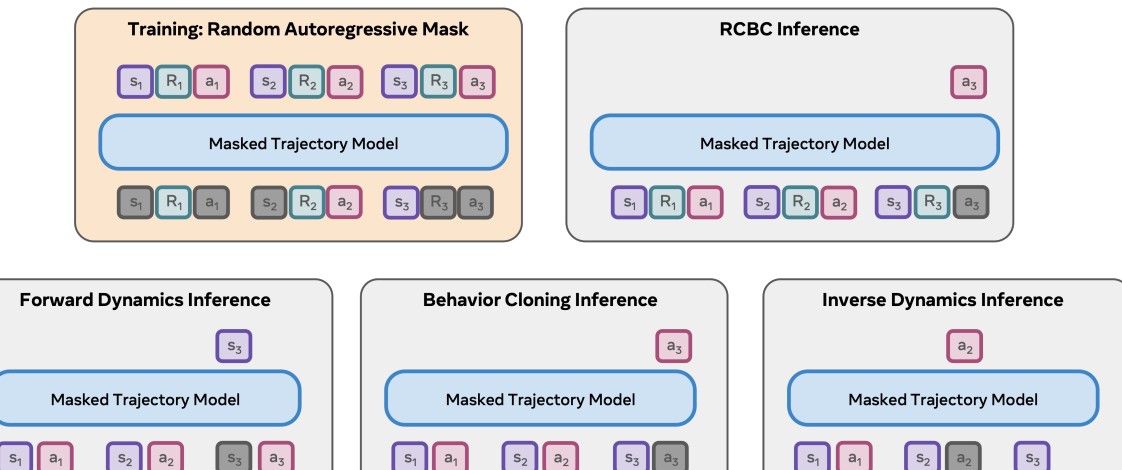

*Figure 3.* **Masking Pattern for Training and Inference.** (Training: box in orange) `MTM` is trained to reconstruct trajectory segments conditioned on a masked view of the same. We use a random autoregressive masking pattern, where elements in the input sequence are randomly masked, with the added constraint that at least one masked token must have no future unmasked tokens. This means the last element in the sequence must necessarily be masked. We note that the input sequence can start and end on arbitrary modalities. In this illustrated example, $R_3$ is the masked token that satisfies the autoregressive constraint. That is the prediction of $R_3$ is conditioned on no future tokens in the sequence. (Inference: boxes in gray) By changing the masking pattern at inference time, `MTM` can either be used directly for offline RL using RCBC (Chen et al., 2021), or be used as a component in traditional RL pipelines as a state representation, dynamics model, policy initialization, and more. These different capabilities are shown in gray. Modes not shown at the input are masked out and modes not shown at the output are not directly relevant for the task of interest.

*Table 1.* **Results on D4RL.** Offline RL results on the V2 locomotion suite of D4RL are reported here, specified by the normalized score as described in Fu et al. (2020). We find that `MTM` outperforms RvS and DT, which also use RCBC for offline RL.

| Environment | Dataset | BC | CQL | IQL | TT | MOPO | RsV | DT | **MTM (Ours)** |
|---|---|---|---|---|---|---|---|---|---|
| HalfCheetah | Medium-Replay | 36.6 | 45.5 | 44.2 | 41.9 | 42.3 | 38.0 | 36.6 | 43.0 |
| Hopper | Medium-Replay | 18.1 | 95.0 | 94.7 | 91.5 | 28.0 | 73.5 | 82.7 | 92.9 |
| Walker2d | Medium-Replay | 26.0 | 77.2 | 73.9 | 82.6 | 17.8 | 60.6 | 66.6 | 77.3 |
| HalfCheetah | Medium | 42.6 | 44.0 | 47.4 | 46.9 | 53.1 | 41.6 | 42.0 | 43.6 |
| Hopper | Medium | 52.9 | 58.5 | 66.3 | 61.1 | 67.5 | 60.2 | 67.6 | 64.1 |
| Walker2d | Medium | 75.3 | 72.5 | 78.3 | 79.0 | 39.0 | 71.7 | 74.0 | 70.4 |
| HalfCheetah | Medium-Expert | 55.2 | 91.6 | 86.7 | 95.0 | 63.7 | 92.2 | 86.8 | 94.7 |
| Hopper | Medium-Expert | 52.5 | 105.4 | 91.5 | 110.0 | 23.7 | 101.7 | 107.6 | 112.4 |
| Walker2d | Medium-Expert | 107.5 | 108.8 | 109.6 | 101.9 | 44.6 | 106.0 | 108.1 | 110.2 |
| Average | | 51.9 | 77.6 | 77.0 | 78.9 | 42.2 | 71.7 | 74.7 | 78.7 |

**D4RL** (Fu et al., 2020) is a popular offline RL benchmark consisting of several environments and datasets. Following a number of prior work, we focus on the locomotion subset: `Walker2D`, `Hopper`, and `HalfCheetah`. For each environment, we consider 4 different dataset settings: `Expert`, `Medium-Expert`, `Medium`, and `Medium-Replay`. The `Expert` dataset is useful for benchmarking imitation learning with BC, while the other datasets enable studying offline RL and other capabilities of `MTM` such as future prediction and inverse dynamics.

**Adroit** (Rajeswaran et al., 2018) is a collection of dexterous manipulation tasks with a simulated five-fingered. We experiment with the `Pen`, and `Door` tasks that test an agent's ability to carefully coordinate a large action-space to accomplish complex robot manipulation tasks. We collect `Medium-Replay` and `Expert` trajectories for each task using a protocol similar to D4RL.

**ExORL** (Yarats et al., 2022) dataset consists of trajectories collected using various unsupervised exploration algorithms. Yarats et al. (2022) showed that TD3 (Fujimoto et al., 2018a) can be effectively used to learn in this benchmark. We use data collected by a ProtoRL agent (Yarats et al., 2021) in the `Walker2D` environment to learn three different tasks: `Stand`, `Walk`, and `Run`.

*Table 2.* **Evaluation of various MTM capabilities.** MTM refers to the model trained with the random autoregressive mask, and evaluated using the appropriate mask at inference time. S-MTM ("Specialized") refers to the model that uses the appropriate mask both during training and inference time. We also compare with a specialized MLP baseline trained separately for each capability. Note that higher is better for BC and RCBC, while lower is better for FD and ID. We find that MTM is often comparable or better than training on specialized masking patterns, or training specialized MLPs. We use a box outline to indicate that a single model was used for all the evaluations within it. The right most column indicates if MTM is comparable or better than S-MTM, and we find this to be true in most cases.

| Domain | Dataset | Task | MLP | S-MTM (Ours) | MTM (Ours) | (MTM) $\gtrsim$ (S-MTM)? |
|---|---|---|---|---|---|---|
| D4RL Hopper | Expert | (↑) BC | $111.14 \pm 0.33$ | $111.81 \pm 0.18$ | $107.35 \pm 7.77$ | ✓ |
| | Expert | (↑) RCBC | $111.17 \pm 0.56$ | $112.64 \pm 0.47$ | $112.49 \pm 0.37$ | ✓ |
| | Expert | (↓) ID | $0.009 \pm 0.000$ | $0.013 \pm 0.000$ | $0.050 \pm 0.026$ | ✗ |
| | Expert | (↓) FD | $0.072 \pm 0.000$ | $0.517 \pm 0.025$ | $0.088 \pm 0.049$ | ✓ |
| D4RL Hopper | Medium Replay | (↑) BC | $35.63 \pm 6.27$ | $36.17 \pm 4.09$ | $29.46 \pm 6.74$ | ✗ |
| | Medium Replay | (↑) RCBC | $88.61 \pm 1.68$ | $93.30 \pm 0.33$ | $92.95 \pm 1.51$ | ✓ |
| | Medium Replay | (↓) ID | $0.240 \pm 0.028$ | $0.219 \pm 0.008$ | $0.534 \pm 0.009$ | ✗ |
| | Medium Replay | (↓) FD | $2.179 \pm 0.052$ | $3.310 \pm 0.425$ | $0.493 \pm 0.030$ | ✓ |
| Adroit Pen | Expert | (↑) BC | $62.75 \pm 1.43$ | $66.28 \pm 3.28$ | $61.25 \pm 5.06$ | ✓ |
| | Expert | (↑) RCBC | $68.41 \pm 2.27$ | $66.29 \pm 1.39$ | $64.81 \pm 1.70$ | ✓ |
| | Expert | (↓) ID | $0.128 \pm 0.001$ | $0.155 \pm 0.001$ | $0.331 \pm 0.049$ | ✗ |
| | Expert | (↓) FD | $0.048 \pm 0.002$ | $0.360 \pm 0.020$ | $0.321 \pm 0.048$ | ✓ |
| Adroit Pen | Medium Replay | (↑) BC | $33.73 \pm 1.00$ | $54.84 \pm 5.08$ | $47.10 \pm 7.13$ | ✗ |
| | Medium Replay | (↑) RCBC | $41.26 \pm 4.99$ | $57.50 \pm 3.76$ | $58.76 \pm 5.63$ | ✓ |
| | Medium Replay | (↓) ID | $0.308 \pm 0.004$ | $0.238 \pm 0.004$ | $0.410 \pm 0.064$ | ✓ |
| | Medium Replay | (↓) FD | $0.657 \pm 0.023$ | $0.915 \pm 0.007$ | $0.925 \pm 0.026$ | ✓ |

## 4.2. Offline RL results

We first test the capability of MTM to learn policies in the standard offline RL setting. To do so, we train MTM with the random autoregressive masking pattern as described in Section 3. Subsequently, we use the Return Conditioned Behavior Cloning (RCBC) mask at inference time for evaluation. This is inspired by DT (Chen et al., 2021) which uses a similar RCBC approach, but with a GPT model.

Our empirical results are presented in Table 1. We find that MTM outperforms the closest algorithms of DT and RvS, suggesting that masked prediction is an effective pre-training task for offline RL when using RCBC inference mask. More surprisingly, MTM is competitive with highly specialized and state-of-the-art offline RL algorithms like CQL (Kumar et al., 2020) and IQL (Kostrikov et al., 2021) despite training with a purely self-supervised learning objective without any explicit RL components.

## 4.3. MTM Capabilities

We next study if MTM is a versatile learner by evaluating it across four different capabilities on Adroit and D4RL datasets. We emphasize that we test these capabilities for a single MTM-model (i.e. same weights) by simply altering the masking pattern during inference time. See Figure 3 for a visual illustration of the inference-time masking patterns.

1. **Behavior Cloning (BC)**: Predict next action given state-action history. This is a standard approach to imitation learning as well as a popular initialization method for subsequent RL (Rajeswaran et al., 2018).

2. **Return Conditioned Behavior Cloning (RCBC)** is similar to BC, but additionally conditions on the desired Return-to-Go. Recent works (Chen et al., 2021; Emmons et al., 2021) have shown that RCBC can lead to successful policies in the offline RL setting.

3. **Inverse Dynamics (ID)**, where we predict the action using the current and future desired state. This can be viewed as a 1-step goal-reaching policy. It has also found application in observation-only imitation learning (Radosavovic et al., 2021; Baker et al., 2022).

4. **Forward Dynamics (FD)**, where we predict the next state given history and current action. Forward dynamics models are an integral component of several model-based RL algorithms (Janner et al., 2019; Rajeswaran et al., 2020; Hafner et al., 2020).

We consider two variations of MTM. The first variant, S-MTM, trains a specialized model for each capability using the corresponding masking pattern at *train time*. The second variant, denoted simply as MTM, trains a single model using the random autoregressive mask specified in Section 3. Subsequently, the same model (i.e. same set of weights) is evaluated for all the four capabilities. We also compare our results with specialized MLP models for each capability. We evaluate the best checkpoint across all models and report mean and standard deviation across 4 seeds, taking the average of 20 trajectory executions per seed. For all

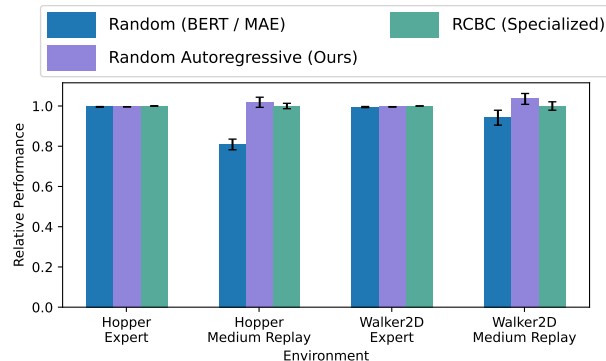

Figure 4. **Impact of Masking Patterns.** This plot shows `MTM` RCBC performance trained with three different masking patterns, random, random autoregressive, and a specialized RCBC mask. We find that autoregressive random often outperforms random, and in most cases is even competitive with the specialized (or oracle) RCBC mask. $Y$-axis normalized with using RCBC mask.

experiments we train on 95% of the dataset and reserve 5% of the data for evaluation. For BC and RCBC results, we report the normalized score obtained during evaluation rollouts. For ID and FD, we report normalized loss values on the aforementioned 5% held-out data.

A snapshot of our results are presented in Table 2 for a subset of environments. Please see Appendix A for detailed results on all the environments. The last column of the table indicates the performance difference between the versatile `MTM` and the specialized `S-MTM`. We find that `MTM` is comparable or even better than specialized masks, and also matches the performance of specialized MLP models. We suspect that specialized masks may require additional tuning of parameters to prevent overfitting or underfitting, whereas random autoregressive masking is more robust across tasks and hyperparameters.

### 4.4. Impact of Masking Patterns

We study if the masking pattern influences the capabilities of the learned model. Figure 4 shows that random autoregressive masking matches or outperforms purely random masking on RCBC for a spread of environments for offline RL. We note that pure random masking, as done in MAE and BERT, which focuses on only learning good representations, can lead to diminished performance for downstream capabilities. Random autoregressive masking mitigates these issues by allowing the learning of a single versatile model while still matching or even exceeding the performance of specialized masks, as seen in Table 2.

### 4.5. Heteromodal Datasets

`MTM` is uniquely capable of learning from heteromodal datasets. This is enabled by the training procedure, where any missing data can be treated as if it were masked. During

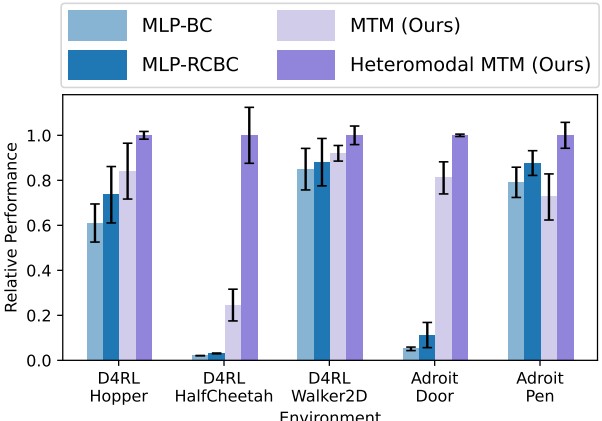

Figure 5. **`MTM` can effectively learn from heteromodal datasets.** Real world data may not always contain action labels. We simulate this setting by training a `MTM` models on Expert datasets across domains where only a small fraction of the data have action labels. Our Heteromodal `MTM` model is able to effectively improve task with the additional data over baseline `MTM` and MLP that train on only the subset of data with actions. $Y$-axis normalized with respect to performance of Heteromodal `MTM`.

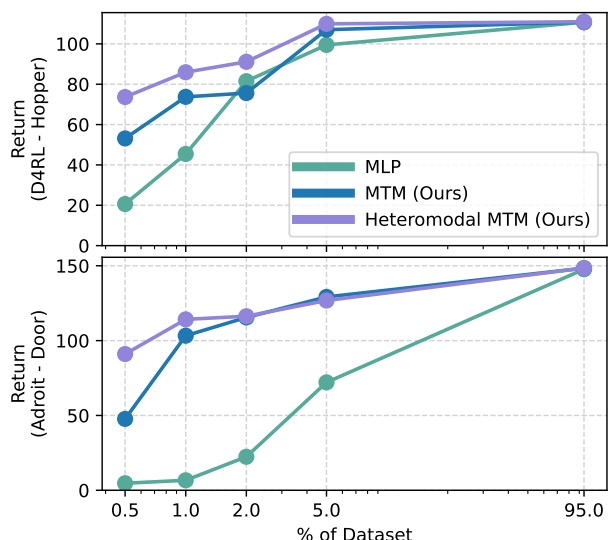

Figure 6. **Dataset efficiency.** We train `MTM` in the D4RL Hopper and Adroit Door environments across a range of dataset sizes, measured by the percent of the original dataset ($\approx 1$ million transitions). We see that `MTM` is able to consistently outperform specialized MLP models in the low data regime. Furthermore, we see that Heteromodal `MTM` (i.e. `MTM` trained on heteromodal data containing both state-only and state-action trajectories) is further able to provide performance improvement in low data regimes.

training we apply the loss only to modes that exist in the dataset. For these experiments we take the `Expert` subset of our trajectory data and remove action labels from the majority of the dataset. The training data consists of 1% of

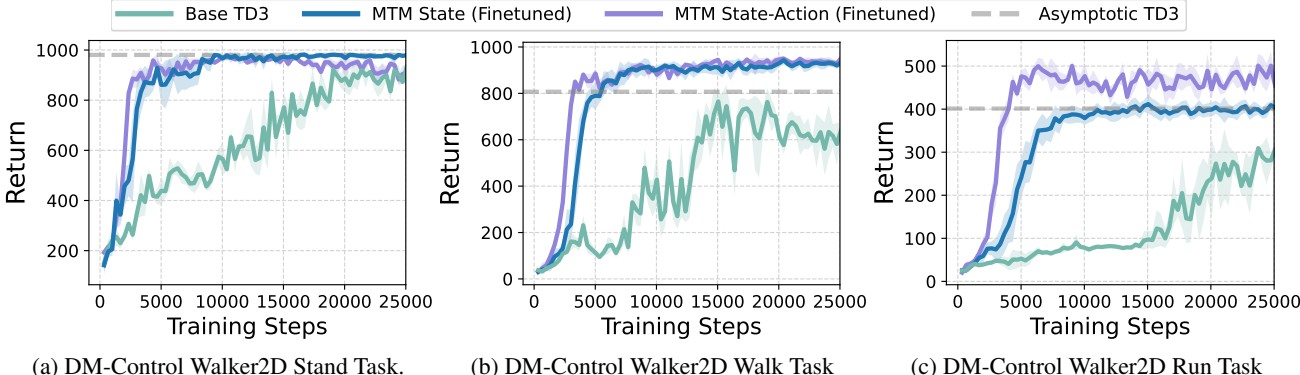

(a) DM-Control Walker2D Stand Task.      (b) DM-Control Walker2D Walk Task      (c) DM-Control Walker2D Run Task

*Figure 7.* **MTM Representations enable faster learning.** The plot visualizes a walker agent's performance as it is trained using TD3 on different representations across 3 tasks (`Stand`, `Walk`, `Run`). The agent is trained completely offline using data from the ExORL dataset. For `MTM` state representations, we encode the raw state with `MTM`. `MTM` state-action representations additionally jointly encode the state and action for the critic of TD3. The learning curves show that finetuned `MTM` representations enable the agent to more quickly learn the task at hand, reaching or exceeding the asymptotic performance of TD3 on raw states. Both `MTM` state representations and `MTM` state-action representations are comparable in terms of learning speed and performance. In addition, we see that in some cases, like the `Run` task, state-action representations from `MTM` helps achieve better performance than alternatives. We also show the asymptotic performance reached by TD3 on raw states and actions after training for 100000 iterations and plot the average of 5 seeds.

the data with all modes (states, actions, return-to-go) and $95\%$ percent of the data with no action labels. As is done in all experiments, the remainder is reserved for testing.

From our initial experiments, we found that naively adding in the state only data during training, and evaluating with the RCBC mask did not always result in improved performance. This was despite improvement in forward dynamics prediction as a result of adding state-only trajectories. Based on this observation, we propose a two-stage action inference procedure. First, we predict future states given current state and desired returns. This can be thought of as a forward dynamics pass where the desired returns are used instead of actions, which are masked out (or more precisely, missing). Next, we predict actions using the current state and predicted future states using the inverse dynamics mask. We refer to this model trained on heteromodal data, along with the two stage inference procedure, as Heteromodal `MTM`. We present the results in Figure 5, where we find that Heteromodal `MTM` consistently improves performance over the baseline MLP and `MTM` that are trained only on the subset of data with action labels.

### 4.6. Data Efficiency

Figure 5 not only showed the effectiveness of `MTM` on heteromodal data, but also that `MTM` is able to achieve higher performance than baseline (specialized) MLPs in the low data regimes. To explicitly test the data efficiency of `MTM`, we study the performance as a function of the training dataset size, and present results in Figure 6. We observe that `MTM` is more sample efficient and achieves higher performance for any given dataset size. Heteromodal `MTM` also outperforms `MTM` throughout, with the performance gap being quite sub-

stantial in the low-data regime. We hypothesize that the data efficiency of `MTM` is due to better usage of the data. Specifically, since the model encounters various masks during training, it must learn general relationships between different elements. As a result, `MTM` may be able to squeeze out more learning signal from any given trajectory.

### 4.7. State Representations of `MTM`

Finally, we study if the representations learned by `MTM` are useful for downstream learning with traditional RL algorithms. If this is the case, `MTM` can also be interpreted as an offline pre-training exercise to help downstream RL. To instantiate this in practice, we consider the setting of offline RL using TD3 on the ExORL dataset. The baseline method is to simply run TD3 on this dataset using the raw state as input to the TD3 algorithm. We compare this to our proposed approach of using `MTM` state representations for TD3. To do this, we first pretrain an `MTM` model on state-action sequences in the ExORL dataset. Subsequently, to use state representations from `MTM`, we simply use the `MTM` encoder to tokenize and encode each state individually. This latent representation of the state can be used in the place of raw states for the TD3 algorithm. The critic of TD3 is conditioned on states and actions. We additionally test state-action representations of `MTM` by using the latent representation of the state and action encoded jointly with `MTM`. We allow end to end finetuning of the representations during training. We compare training TD3 on raw states to training TD3 with (a) state representations from the `MTM` model, and (b) state-action representations from the `MTM` model with the offline RL loss (i.e. TD3 objective).

Figure 7 depicts the learning curves for the aforementioned

experiment. In all cases we see significant improvement in training efficiency by using `MTM` representations – both with state and state-action representations. In the `Walk` task, we note it actually *improves* over the asymptotic performance of the base TD3 (Fujimoto et al., 2018a) algorithm within 10% of training budget. Additionally, we find that the state-action representation from `MTM` can provide significant benefits, as in the case of the `Walk` task. Here, finetuning state-action representation from `MTM` leads to better asymptotic performance compared to state-only representation or learning from scratch. We provide additional plots of `MTM` frozen representations in Appendix E.3

## 5. Summary

In this paper, we introduced `MTM` as a versatile and effective approach for sequential decision making. We empirically evaluated the performance of `MTM` on a variety of continuous control tasks and found that a single pretrained model (i.e. same weights) can be used for different downstream purposes like inverse dynamics, forward dynamics, imitation learning, offline RL, and representation learning. This is accomplished by simply changing the masks used at inference time. In addition, we showcase how `MTM` enables training on heterogeneous datasets without any change to the algorithm. Future work includes incorporating training in online learning algorithms for more sample efficient learning, scaling `MTM` to longer trajectory sequences, and more complex modalities like videos.

## Acknowledgements

The authors thank researchers and students in Meta AI and Berkeley Robot Learning Lab for valuable discussions. Philipp Wu was supported in part by the NSF Graduate Research Fellowship Program. Arjun Majumdar was supported in part by ONR YIP and ARO PECASE. The views and conclusions contained herein are those of the authors and should not be interpreted as necessarily representing the official policies or endorsements, either expressed or implied, of the U.S. Government, any sponsor, or employer.

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

# A. Additional MTM Results

*Table A.1.* Evaluation of MTM capabilities on D4RL.

| Domain | Dataset | Task | MLP | S-MTM (Ours) | MTM (Ours) |
|---|---|---|---|---|---|
| D4RL Hopper | Expert | BC | $111.14 \pm 0.33$ | $111.81 \pm 0.18$ | $107.35 \pm 7.77$ |
| | Expert | RCBC | $111.17 \pm 0.56$ | $112.64 \pm 0.47$ | $112.49 \pm 0.37$ |
| | Expert | ID | $0.009 \pm 0.000$ | $0.013 \pm 0.000$ | $0.050 \pm 0.026$ |
| | Expert | FD | $0.072 \pm 0.000$ | $0.517 \pm 0.025$ | $0.088 \pm 0.049$ |
| D4RL Hopper | Medium Expert | BC | $58.75 \pm 3.79$ | $60.85 \pm 3.14$ | $54.96 \pm 2.44$ |
| | Medium Expert | RCBC | $110.22 \pm 0.99$ | $113.00 \pm 0.39$ | $112.41 \pm 0.23$ |
| | Medium Expert | ID | $0.015 \pm 0.000$ | $0.015 \pm 0.001$ | $0.053 \pm 0.003$ |
| | Medium Expert | FD | $0.139 \pm 0.001$ | $0.938 \pm 0.062$ | $0.077 \pm 0.005$ |
| D4RL Hopper | Medium | BC | $55.93 \pm 1.12$ | $56.74 \pm 0.56$ | $57.64 \pm 3.37$ |
| | Medium | RCBC | $62.20 \pm 3.41$ | $69.20 \pm 1.60$ | $70.48 \pm 4.62$ |
| | Medium | ID | $0.022 \pm 0.001$ | $0.030 \pm 0.001$ | $0.143 \pm 0.035$ |
| | Medium | FD | $0.153 \pm 0.002$ | $1.044 \pm 0.061$ | $0.206 \pm 0.064$ |
| D4RL Hopper | Medium Replay | BC | $35.63 \pm 6.27$ | $36.17 \pm 4.09$ | $29.46 \pm 6.74$ |
| | Medium Replay | RCBC | $88.61 \pm 1.68$ | $93.30 \pm 0.33$ | $92.95 \pm 1.51$ |
| | Medium Replay | ID | $0.240 \pm 0.028$ | $0.219 \pm 0.008$ | $0.534 \pm 0.009$ |
| | Medium Replay | FD | $2.179 \pm 0.052$ | $3.310 \pm 0.425$ | $0.493 \pm 0.030$ |
| D4RL Walker2D | Expert | BC | $109.28 \pm 0.12$ | $108.76 \pm 0.32$ | $107.08 \pm 1.47$ |
| | Expert | RCBC | $112.21 \pm 0.31$ | $109.83 \pm 0.58$ | $110.08 \pm 0.82$ |
| | Expert | ID | $0.021 \pm 0.000$ | $0.055 \pm 0.001$ | $0.233 \pm 0.038$ |
| | Expert | FD | $0.077 \pm 0.001$ | $0.233 \pm 0.012$ | $0.177 \pm 0.031$ |
| D4RL Walker2D | Medium Expert | BC | $108.45 \pm 0.31$ | $108.49 \pm 1.00$ | $75.64 \pm 7.78$ |
| | Medium Expert | RCBC | $110.47 \pm 0.38$ | $110.43 \pm 0.30$ | $110.21 \pm 0.31$ |
| | Medium Expert | ID | $0.019 \pm 0.000$ | $0.038 \pm 0.001$ | $0.213 \pm 0.030$ |
| | Medium Expert | FD | $0.088 \pm 0.001$ | $0.221 \pm 0.013$ | $0.167 \pm 0.032$ |
| D4RL Walker2D | Medium | BC | $75.91 \pm 1.87$ | $75.87 \pm 0.44$ | $59.82 \pm 7.06$ |
| | Medium | RCBC | $78.76 \pm 2.26$ | $78.64 \pm 2.05$ | $78.08 \pm 2.04$ |
| | Medium | ID | $0.026 \pm 0.001$ | $0.055 \pm 0.002$ | $0.214 \pm 0.145$ |
| | Medium | FD | $0.116 \pm 0.002$ | $0.236 \pm 0.012$ | $0.175 \pm 0.162$ |
| D4RL Walker2D | Medium Replay | BC | $23.39 \pm 2.75$ | $48.45 \pm 2.84$ | $21.98 \pm 2.77$ |
| | Medium Replay | RCBC | $72.85 \pm 5.23$ | $78.33 \pm 2.11$ | $77.32 \pm 1.79$ |
| | Medium Replay | ID | $0.532 \pm 0.017$ | $0.493 \pm 0.018$ | $0.921 \pm 0.032$ |
| | Medium Replay | FD | $1.224 \pm 0.011$ | $0.883 \pm 0.011$ | $0.446 \pm 0.016$ |
| D4RL HalfCheetah | Expert | BC | $93.14 \pm 0.16$ | $95.21 \pm 0.44$ | $94.19 \pm 0.21$ |
| | Expert | RCBC | $94.16 \pm 0.35$ | $95.12 \pm 0.64$ | $94.83 \pm 0.72$ |
| | Expert | ID | $0.001 \pm 0.000$ | $0.003 \pm 0.000$ | $0.009 \pm 0.001$ |
| | Expert | FD | $0.009 \pm 0.000$ | $0.018 \pm 0.003$ | $0.005 \pm 0.001$ |
| D4RL HalfCheetah | Medium Expert | BC | $68.04 \pm 1.57$ | $77.88 \pm 7.21$ | $65.73 \pm 5.69$ |
| | Medium Expert | RCBC | $93.49 \pm 0.29$ | $94.85 \pm 0.32$ | $94.78 \pm 0.39$ |
| | Medium Expert | ID | $0.001 \pm 0.000$ | $0.001 \pm 0.000$ | $0.012 \pm 0.002$ |
| | Medium Expert | FD | $0.014 \pm 0.000$ | $0.043 \pm 0.008$ | $0.009 \pm 0.001$ |
| D4RL HalfCheetah | Medium | BC | $42.87 \pm 0.11$ | $43.37 \pm 0.14$ | $43.19 \pm 0.34$ |
| | Medium | RCBC | $44.43 \pm 0.26$ | $43.83 \pm 0.22$ | $43.65 \pm 0.08$ |
| | Medium | ID | $0.001 \pm 0.000$ | $0.005 \pm 0.000$ | $0.027 \pm 0.017$ |
| | Medium | FD | $0.020 \pm 0.000$ | $0.053 \pm 0.011$ | $0.020 \pm 0.010$ |
| D4RL HalfCheetah | Medium Replay | BC | $36.81 \pm 0.52$ | $39.03 \pm 0.78$ | $19.64 \pm 11.26$ |
| | Medium Replay | RCBC | $40.55 \pm 0.18$ | $42.94 \pm 0.33$ | $43.08 \pm 0.43$ |
| | Medium Replay | ID | $0.003 \pm 0.000$ | $0.005 \pm 0.000$ | $0.036 \pm 0.012$ |
| | Medium Replay | FD | $0.059 \pm 0.000$ | $0.058 \pm 0.010$ | $0.028 \pm 0.007$ |

Table A.2. Evaluation of `MTM` capabilities on Adroit.

| Domain | Dataset | Task | MLP | S-MTM (Ours) | MTM (Ours) |
|---|---|---|---|---|---|
| Adroit Pen | Expert | BC | $62.75 \pm 1.43$ | $66.28 \pm 3.28$ | $61.25 \pm 5.06$ |
| | Expert | RCBC | $68.41 \pm 2.27$ | $66.29 \pm 1.39$ | $64.81 \pm 1.70$ |
| | Expert | ID | $0.128 \pm 0.001$ | $0.155 \pm 0.001$ | $0.331 \pm 0.049$ |
| | Expert | FD | $0.048 \pm 0.002$ | $0.360 \pm 0.020$ | $0.321 \pm 0.048$ |
| Adroit Pen | Medium Replay | BC | $33.73 \pm 1.00$ | $54.84 \pm 5.08$ | $47.10 \pm 7.13$ |
| | Medium Replay | RCBC | $41.26 \pm 4.99$ | $57.50 \pm 3.76$ | $58.76 \pm 5.63$ |
| | Medium Replay | ID | $0.308 \pm 0.004$ | $0.238 \pm 0.004$ | $0.410 \pm 0.064$ |
| | Medium Replay | FD | $0.657 \pm 0.023$ | $0.915 \pm 0.007$ | $0.925 \pm 0.026$ |
| Adroit Door | Expert | BC | $147.68 \pm 0.25$ | $149.46 \pm 0.29$ | $149.19 \pm 0.72$ |
| | Expert | RCBC | $148.81 \pm 0.32$ | $150.50 \pm 0.14$ | $149.93 \pm 0.19$ |
| | Expert | ID | $0.385 \pm 0.001$ | $0.427 \pm 0.003$ | $0.484 \pm 0.024$ |
| | Expert | FD | $0.199 \pm 0.011$ | $0.541 \pm 0.020$ | $0.618 \pm 0.210$ |
| Adroit Door | Medium Replay | BC | $27.75 \pm 5.03$ | $49.24 \pm 26.85$ | $16.30 \pm 10.10$ |
| | Medium Replay | RCBC | $71.51 \pm 8.62$ | $75.41 \pm 8.20$ | $51.92 \pm 9.13$ |
| | Medium Replay | ID | $0.532 \pm 0.001$ | $0.589 \pm 0.005$ | $0.629 \pm 0.014$ |
| | Medium Replay | FD | $0.976 \pm 0.033$ | $2.225 \pm 0.061$ | $2.251 \pm 0.230$ |

# B. Additional Environment Details

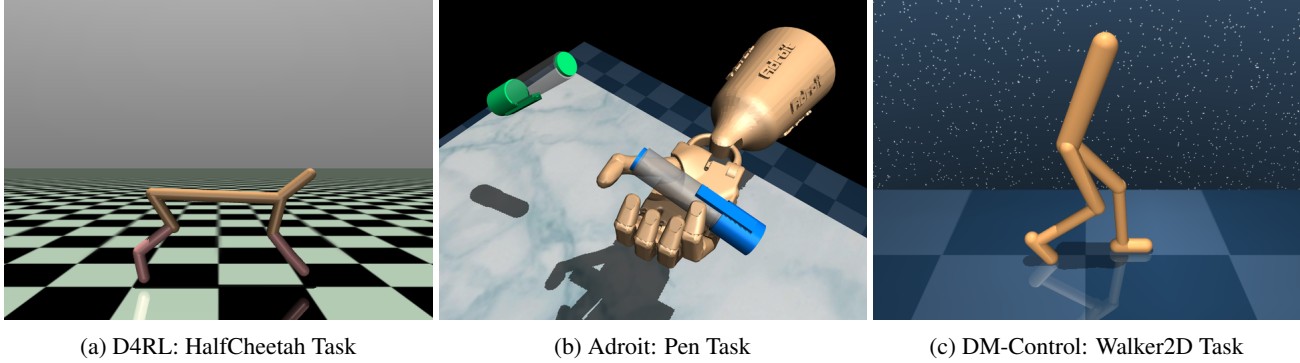

(a) D4RL: HalfCheetah Task   (b) Adroit: Pen Task   (c) DM-Control: Walker2D Task

Figure B.1. **Continues Control Evaluation Settings.**

Here we provide additional details on each experiment setting. In general, our empirical evaluations are based on the standard versions of D4RL, Adroit, and ExORL. These benchmarks and setups are widely used in the community for studying various aspects of offline learning. The raw state space provided by these benchmarks typically comprise a mix of positions and velocities of different joints, bodies, and objects in the environment. We preprocess each dataset by normalizing the data before training.

**D4RL** (Fu et al., 2020) is a popular offline RL benchmark. As mentioned in Section 4.1, we test `MTM` on the locomotion suit of D4RL. The locomotion suite uses the `Walker`, `Hopper`, and `HalfCheetah` environments provided by OpenAI Gym (Brockman et al., 2016). We consider 4 different dataset settings: `Expert`, `Medium-Expert`, `Medium`, and `Medium-Replay`. These datasets are collected by taking trajectories of a SAC (Haarnoja et al., 2017) agent at various points in training.

**Adroit** (Rajeswaran et al., 2018) is a collection of dexterous manipulation tasks with a simulated five-fingered. Our `MTM` experiments use the `Pen`, and `Door` tasks. To match the setup of D4RL, we collect `Medium-Replay` and `Expert` trajectories for each task. This is done by training an expert policy. The `Expert` dataset comprises of rollouts of the converged policy with a small amount of action noise. The `Medium-Replay` dataset is a collection of trajectory rollouts from various checkpoints during training of the expert policy, before policy convergence. The original Adroit environment provides a dense reward and a sparse measure of task completion. For `MTM` experiments, we use the task completion signal

as an alternative to reward, which provides a more grounded signal of task performance (a measure of the number of time steps in the episode where the task is complete).

**ExORL** (Yarats et al., 2022) dataset consists of trajectories collected using various unsupervised exploration algorithms. ExORL leverages dm_control developed by Tunyasuvunakool et al. (2020). We use data collected by a ProtoRL agent (Yarats et al., 2021) in the Walker2D environment to evaluate the effectiveness of `MTM` representations on three different tasks: `Stand`, `Walk`, and `Run`. As the pretraining dataset has not extrinsic reward, `MTM` is trained with only states and actions. During downstream TD3 learning, all trajectories are relabeled with the task reward.

## C. Model and Training Details

### C.1. MLP Baseline Hyperparameters

*Table C.1.* MLP Hyperparameters

| | Hyperparameter | Value |
|---|---|---|
| **MLP** | | |
| | Nonlinearity | GELU |
| | Batch Size | 4096 |
| | Embedding Dim | 1024 |
| | # of Layers | 2 |
| **Adam Optimizer** | | |
| | Learning Rate | 0.0002 |
| | Weight Decay | 0.005 |
| | Warmup Steps | 5000 |
| | Training Steps | 140000 |
| | Scheduler | cosine decay |

### C.2. `MTM` Model Hyperparameters

*Table C.2.* `MTM` Hyperparameters

| | Hyperparameter | Value |
|---|---|---|
| **General** | | |
| | Nonlinearity | GELU |
| | Batch Size | 1024 |
| | Trajectory-Segment Length | 4 |
| | Scheduler | cosine decay |
| | Warmup Steps | 40000 |
| | Training Steps | 140000 |
| | Dropout | 0.10 |
| | Learning Rate | 0.0001 |
| | Weight Decay | 0.01 |
| **Bidirectional Transformer** | | |
| | # of Encoder Layers | 2 |
| | # Decoder Layers | 1 |
| | # Heads | 4 |
| | Embedding Dim | 512 |
| **Mode Decoding Head** | | |
| | Number of Layers | 2 |
| | Embedding Dim | 512 |

### C.3. `MTM` Training Details

In this section, we specify additional details of `MTM` for reproduction. Numerical values of the hyperparamters are found in table C.1. The architecture follows the structure of (He et al., 2021) and (Liu et al., 2022), which involves a bidirectional transformer encoder and a bidirectional transformer decoder. For each input modality there is a learned projection into the embedding space. In addition we add a 1D sinusoidal encoding to provide time index information. The encoder only processes unmasked tokens. The decoder processes the full trajectory sequence, replacing the masked out tokens with mode specific mask tokens. At the output of the decoder, we use a 2 Layer MLP with Layer Norm (Ba et al., 2016). For training the model we use the AdamW optimizer (Kingma & Ba, 2017; Loshchilov & Hutter, 2019) with a warm up period and cosine learning rate decay.

As we rely on the generative capabilities of `MTM` which can be conditioned on a variety of different input tokens at inference time, we train `MTM` with a range of mask ratios that are randomly sampled. We use a range between 0.0 and 0.6. Our random autoregressive masking scheme also requires that at least one token is predicted without future context. This is done by randomly sampling a time step and token, and masking out future tokens.

## D. Effect of training trajectory length

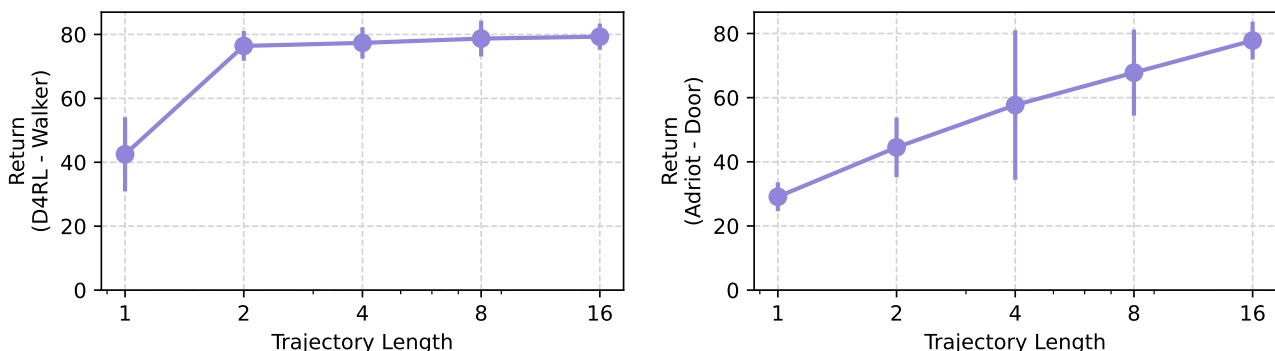

*Figure D.1.* **Effect of Trajectory Training Length.** This plot depicts the effect of changing the training trajectory length on RCBC performance, all other hyperparameters held constant. The left side shows the performance of D4RL `Walker2D` and the right on Adroit `Door`, both using their corresponding `Medium-Replay` Dataset.

Figure D.1 illustrates the effect of training trajectory length on performance. We observe that increased trajectory length has benefits in training performance. We hypothesize that with longer trajectory lengths, `MTM` is able to provide richer training objectives, as the model now must learn how to predict any missing component of a longer trajectory. This is especially apparent in the Adroit `Door` task, where we see RCBC performance increasing strongly with trajectory training length. This suggests that better results could be achieved with longer horizon models. We see that this benefit provides diminishing returns for much longer trajectories (and additionally increases training time), which is most apparent in the D4RL `Walker2D` task. However, for practicality, we fix the trajectory length to 4 for all other experiments, and tune hyperparameters for this trajectory training length. Factors such as mask ratio could be tuned to optimize performance and training time for longer trajectory lengths, but we leave this exploration for future work.

# E. Additional plots

## E.1. Masking Patterns

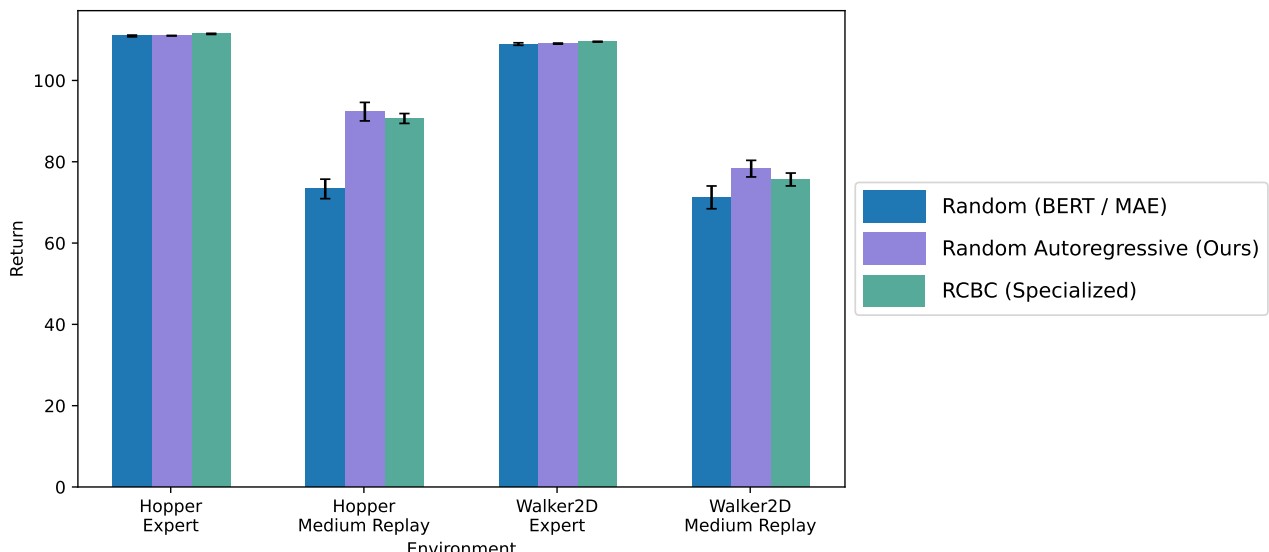

*Figure E.1.* **Impact of Masking Patterns.** This plot shows MTM RCBC performance trained with three different masking patterns, random, random autoregressive, and a specialized RCBC mask. This is a repeat of Figure 4, except the $Y$-axis is unscaled.

## E.2. Heteromodal MTM

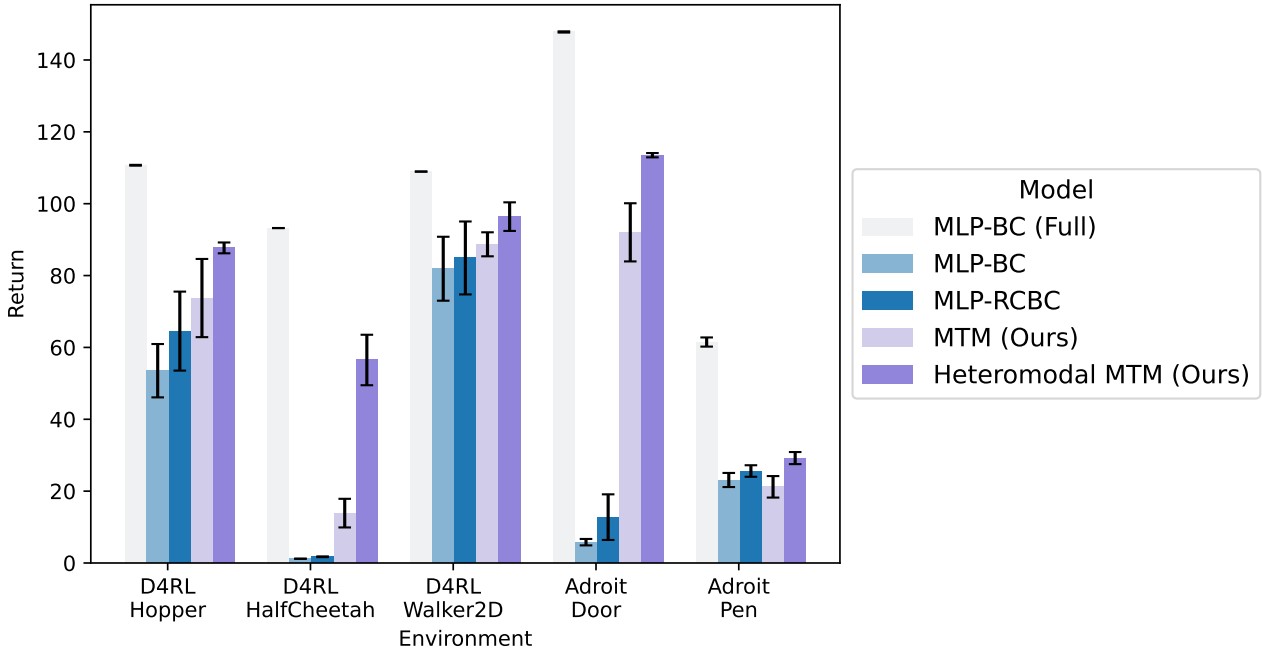

*Figure E.2.* **MTM can effectively learn from heteromodal datasets.** This figure, which shows the performance of our Heteromodal MTM model, is a repeat of Figure 5, except the $Y$-axis is unscaled. Instead we observe the absolute return for each environment. In addition we provide the performance of BC trained on the entire training set (95% of the provided dataset) as reference for the **oracle** performance that can be achieved.

## E.3. Representation Learning

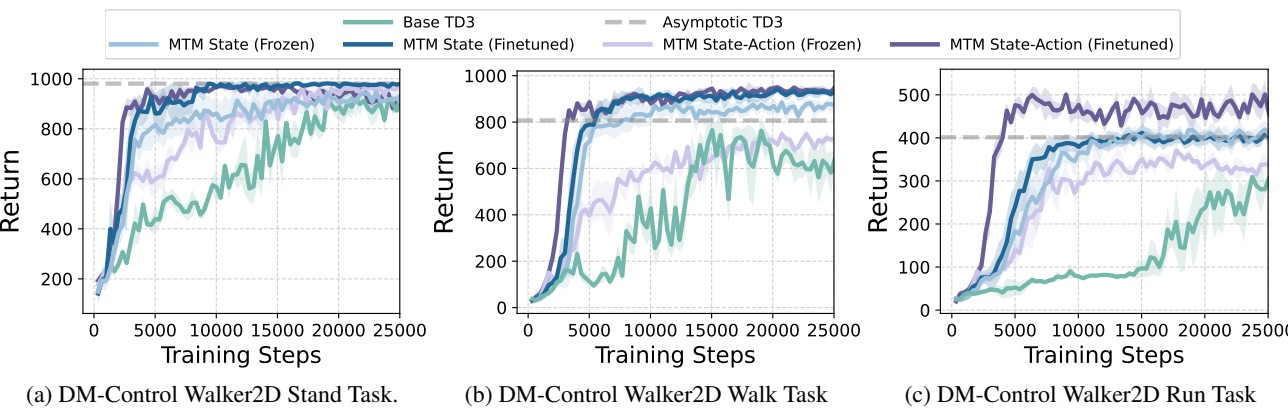

(a) DM-Control Walker2D Stand Task.     (b) DM-Control Walker2D Walk Task     (c) DM-Control Walker2D Run Task

*Figure E.3.* **Finetuned and frozen MTM representations.** Here we additionally provide the learning curves for frozen MTM representations on top of those provided in Figure 7. Both frozen MTM features and finetuned MTM features enable faster learning, but we do see that finetuning offers the best learning benefits across tasks.