# OpenReview forum: "Masked Trajectory Models for Prediction, Representation, and Control"
_ICLR.cc/2023/Workshop/RRL — RRL 2023 Poster_

### Official Review · Reviewer_kdYE · 2023-02-26
**Thorough evaluation of representation learning with masking**

**Rating:** 4
**Confidence:** 4

**Review:**

The paper explores the question of whether a successful idea in vision and language - of learning representations with masking can also extend to the reinforcement learning setting. To test this they combine masking with the transformers to predict masked trajectory sequences from the rest of the trajectory. They utilize (state-action-returns) as each element of the trajectory. This model allows them to solve a number of problems in RL in a unified way - Forward dynamics, inverse dynamics, BC, and offline RL.

Strengths:
1. With a variety of experiments authors successfully show that the method if able to achieve improved performance across all tasks using a single model - showing the importance of features learned when training using masking.
2. The authors present an ablation of important components of their method which is a good addition.

Questions:
1. In the purely offline setting with downstream representations, should an offline RL algorithm have been compared instead of TD3 since it is not clear if the vanilla method is failing due to lack of good representations or the overestimation issue.

---

### Official Review · Reviewer_EACQ · 2023-02-28
**A well structured paper that presents a fairly novel way of training a transformer model using random trajectory masking such that the learnt model obtains several capabilities that are potentially useful to RRL**

**Rating:** 3
**Confidence:** 3

**Review:**

## Summary

The authors present the Masked Trajectory Model (MTM) – a transformer model trained using a novel random trajectory-masking self-supervised training objective. The authors demonstrate the versatile capabilities exhibited by the MTM, many of which are relevant to Reincarnating RL (RRL).

## Review

The authors make a fairly novel contribution towards leveraging transformers and self-supervised learning in RL, which I believe is indeed relevant to the RRL workshop. I believe the results presented by the authors to substantiate their argument are significant and sound, and therefore I recommend an *accept* for this paper to the RRL workshop.

## Overall Score: 3 ⭐⭐⭐

Accept.

## Marking Rubric

### Relevance - 3/5 ⭐⭐⭐

The MTM presented by the authors demonstrates several capabilities which are relevant to RRL, including:

- stand-alone offline RL algorithm (in the form of return-conditioned behaviour cloning)
- pre-trained state representation that accelerates the learning of traditional RL algorithms
- policy initialisation through behaviour cloning
- pre-trained world model for model-based RL algorithms

Together the MTM’s capabilities highlight the versatility of the transformer model, trained using an appropriate self-supervised  for RRL and it is our belief that the work is likely to catalyse interesting discussions at the RRL workshop.

I believe the work could have further improved upon its *********relevance********* to the workshop by including a section that explicitly discusses RRL and how MTM model relates to it.

### Novelty - 3/5 ⭐⭐⭐

The MTM builds on related works that use transformer models and self-supervised learning for RL. The primary contribution of this work is to demonstrate that a transformer model trained with a novel random trajectory-masking self-supervised training objective, demonstrates a diverse set of capabilities. I believe that the novel training objective is a fairly noteworthy contribution, well-supported by the results presented in the paper.

### Significance / Importance - 3/5 ⭐⭐⭐

I feel the results presented in this work are likely to be fairly important for the RRL community since it demonstrates transformer models trained with the appropriate self-supervised training objective can be versatile tools for  RRL. On the other hand, I feel the paper could have benefited from having a section on discussing relevance and significance for RRL more explicitly and not left it to the reader to infer themselves. This would have helped bolster the *relevance* and *significance* scores on this paper for the RRL workshop.

### Soundness - 3/5 ⭐⭐⭐

The authors ran 4 independent seeds for each experiment, which is a reasonable amount for a workshop paper but the authors should do more for main track papers. The authors also included uncertainty estimates on most of their results, except in table 1 and figure 6.

### Scholarship - 3/5 ⭐⭐⭐

The authors covered the related work in a good amount of detail. The authors should however pay more attention to the quality of their list of references. Authors should take care to cite versions of papers in conferences and journals instead of the ArXiv versions. For example, the authors cited the ArXiv version of  ***Conservative Q-Learning (Kumar et al.),*** despite it having been published at NeurIPS.

### Clarity - 4/5 ⭐⭐⭐⭐

Over all, the paper was well structured and well written. Furthermore,  the tables and figures where used effectively to communicate the main ideas and results of the paper.

### Reproducibility - 3/5 ⭐⭐⭐

The authors provide hyper-parameter details in the appendix as well as model architecture details. However, they do not provide any code, which would bolster their reproducibility score.